# Factors associated with unmet need for limiting childbirth among women living with HIV in Togo: An averaging approach

Issifou Yaya[1,2], Aboubakari Nambiema[2], Sokhna Dieng[3], Lihanimpo Djalogue[4], Mawuényégan Kouamivi Agboyibor[5], Mathias Kouamé N'Dri[6], Takiyatou Baba-Toherou[2], Akouda Akessiwe Patassi[7], Dadja Essoya Landoh[8], Kanfitine Kolani[9], Abdoul-Samadou Aboubakari[10], Bayaki Saka[11] *

1 IRD, INSERM, Univ Montpellier, TransVIHMI, Montpellier, France, 2 Association des Chercheurs Africains en Santé des Populations (ACASP), Paris, France, 3 Aix Marseille Univ, INSERM, IRD, SESSTIM, Sciences Economiques & Sociales de la Santé & Traitement de l'Information Médicale, Marseille, France, 4 Service de Médecine Interne, CHU Kara, Kara, Togo, 5 Espoir Vie Togo, Lomé, Togo, 6 Département Épidémiologie-Recherche Clinique, Unité d'Épidémiologie, Institut Pasteur de Côte d'Ivoire, Abidjan, Côte d'Ivoire, 7 Service de Maladies Infectieuses, CHU Sylvanus Olympio, Université de Lomé, Lomé, Togo, 8 World Health Organization, Country Office of Togo, Lomé, Togo, 9 Service de Gynécologie-Obstétrique, Clinique Biasa, Lomé, Togo, 10 Service de Gynéco-Obstétrique, CHU Kara, Kara, Togo, 11 Service de Dermatologie, CHU Sylvanus Olympio, Lomé, Togo

* barthelemysaka@yahoo.fr

**Data Availability Statement:** All relevant data are within the paper and its Supporting Information files.

## Abstract

### Background

Access to antiretroviral treatment has improved the life expectancy of HIV-positive patients, most often associated with a desire to limit childbearing. Women living with HIV (WLHIV) commonly have unmet need for contraception and could be at risk of unintended pregnancy. Preventing unintended pregnancies among women living with HIV are effective strategies to eliminate mother-to-child transmission of HIV.

### Objective

The aim of this study was to assess unmet need for limiting childbirth and its associated factors among women living with HIV in Togo.

### Methods

This facility based cross-sectional study was conducted, between June and August 2016, among WLHIV in their reproductive age (15–49 years) in HIV-care settings in Centrale and Kara regions Data was collected using a structured and pretested questionnaire. WLHIV who desired to limit childbirth but not using contraception were considered to have unmet need of birth limitations. Univariate and multivariate Poisson regression models with robust variance were performed to identify associated factors with unmet needs. A multi-model averaging approach was used to estimate the degree of the association between these factors and the unmet need of birth limitations.

**Funding:** The author(s) received no specific funding for this work.

**Competing interests:** The authors have declared that no competing interests exist.

**Abbreviations:** HIV, Human immunodeficiency virus; AIDS, acquired immunodeficiency syndrome; aPR, adjusted prevalence ratio; ART, Antiretroviral therapy; PLWHA, People Living With HIV and AIDS; WLHIV, women living with HIV; FP, Family planning; MTCT, mother-to-child transmission.

## Results

A total of 443 WLHIV were enrolled, with mean age of 34.5 years (standard deviation [SD] = 7.0). Among them 244 (55.1%) were in couple and 200 (45.1%) had at least the secondary level of education. 39.1% were followed-up in a private healthcare facility. At the time of the survey, 40.0% did not desire childbearing but only 9.0% (95% CI [6.7–12.1]) of them expressed unmet needs for limiting childbirth. In multivariable analysis, associated factors with unmet needs of birth limitations were: being aged 35 years or more (adjusted prevalence ratio (aPR) = 3.11, 95% confidence intervals (95% CI) [1.52–6.38]), living in couple (aPR = 2.32 [1.15–4.65]), living in Kara region (aPR = 0.10 [0.01–0.76]), being followed in a private healthcare facility (aPR = 0.08[0.01–0.53]) and having severe HIV symptoms (aPR = 3.50 [1.31–9.37]).

## Conclusion

Even though the unmet need for births limitation was relatively low among WLHIV in Togo, interventions to improve more access to contraceptive methods, and targeting 35 to 49 years old women, those in couple or followed in the public healthcare facilities would contribute to the eradication of mother-to-child transmission of HIV.

## Background

Improving the reproductive health of women living with HIV (WLHIV), including access to high-quality services for family planning would help slow the HIV epidemic in low- and middle-income countries [1]. In low-income countries women are disproportionately affected by HIV [2,3]. For every three new HIV infections among young men aged 15–24 years in Western and Central Africa there were five new infections among young women [3,4]. It is estimated that women account for nearly two-thirds of the people living with HIV/AIDS (PLWHA) in this region.

WLHIV of reproductive age might continue to spread the HIV epidemic through the transmission of HIV to their children during pregnancy or breastfeeding. For HIV-positive pregnant women, in the absence of any intervention related to HIV prevention, the risk of HIV transmission from mother to child is estimated at 15%–30% during pregnancy and childbirth, and 10% to 20% during breast-feeding [5–7]. In 2016, about 87% (140,000 cases) of new HIV infections worldwide among children occurred, the vast majority (over 90%) through mother-to-child transmission (MTCT) during pregnancy, delivery or breast feeding periods [1], in 23 sub-Saharan African countries, where a 48% decline in new child infections since 2010 was reported [8].

Prevention of MTCT of HIV, advocated by WHO as one of the most effective strategies in the fight against HIV/AIDS, is receiving increasing attention at the international, regional and national levels. Access to antiretroviral treatment (ART) and preventing unintended pregnancies among WLHIV are effective strategies to eliminate MTCT [8,9].

In Western and Central Africa, it was estimated 330,000 the number of pregnant women living with HIV in 2017, of whom less than half (42%) received ART to prevent mother-to-child transmission of HIV. Consequently, the rate of mother-to-child transmission (including during the breastfeeding period) was 20.2% [10]. Family planning (FP), by preventing unintended pregnancies and helping to space childbirth for women and couples living with HIV,

constitute an important measures to prevent MTCT of HIV [11,12]. Several studies have shown low prevalence of contraceptive use among HIV-positive compared to their HIV-negative counterparts [13,14], and therefore they have higher unmet need for FP and reproductive health services [15,16], and higher level of unplanned pregnancy [17,18]. The unmet need for FP is increasingly a reliable indicator of reproductive health among WLHIV [11,19,20]. The level of unmet need in FP is the result of several factors, including individual, contextual or structural factors. In low-income countries, availability, cost and geographic accessibility are major barriers to contraceptive use in women of reproductive age, including WLHIV. In a DHS-based study addressing unmet need for FP in WLHIV in Lesotho, Adair [21] found that currently HIV-positive married women, or older WLHIV, were more likely to reported unmet need for FP, while WLHIV with higher household wealth reported lower level of unmet need for FP. But what about the prevalence of unmet need for FP in WLHIV in west Africa, particularly in Togo? And how can we explain it among this group? Even if few studies have examined the use of modern contraception in WLHIV, little is known about unmet need for limiting childbirth among WLHIV in Togo.

A previous study among WLHIV of childbearing age in Togo found a prevalence of 73.1% for the use of modern contraceptives (74.7% for condom use alone and 16.9% in combination with hormonal contraceptive) [22]. It was also reported in that study the education level, marital status, WHO clinical stage, follow-up in private care facilities and having a child were associated factors modern contraception use [22].

This study aimed to assess the prevalence and identify associated factors of unmet need for limiting childbirth among WLHIV attending HIV clinics in two regions of Togo.

## Methods

### Study design

Between May and August 2016, a cross-sectional study was conducted among WLHIV of reproductive age (15–49 years) and sexually active attending HIV clinics in two health regions of Togo including the Centrale and Kara regions.

### Setting and study population

More than 30 medical clinics were accredited to deliver HIV-related care in the Centrale and Kara regions, two of the Togo's six health regions located about 350 km and 420 km from the capital city Lomé, respectively. About 10,361 PLWHA, including 616 children were followed-up in these two health regions [23]. The prevalence of HIV infection in these two regions was 2.2% in the Centrale region and 1.8% in the Kara region in 2013 [24]. The both health regions are characterized by important population mobility seasonally, mainly from the surrounding health regions, and sometimes from outside of the country, occurring during the cultural and traditional events that could increase the risks of HIV transmission.

WLHIV aged 15–49 years who reported a sexual partner in the six months prior the study and who were followed-up at selected study's sites for HIV care or active ART (for at least 6 months) were included in this study. But women who were pregnant at the moment of the study were excluded.

### Sampling

The sampling procedure was described in a previous study [22]. In order to ensure representative HIV clinics with a high number of PLWHA, we performed a random probability sampling proportional to the number of patients in the active file of the 30 HIV clinics in the two

regions. We assumed that a sample of 15% of HIV clinics should be representative of all the HIV clinics. This led to the random selection of 5 HIV clinics for the implementation of this study. "*Then, a non-probabilistic, convenience sampling was performed*" [22]. In the selected HIV clinics, it was proposed to participate to this study, any PLWHA who were admitted in the active file for a follow-up from May to August 2016, who met the inclusion criteria and who signed the consent form to participate in the study. "The prevalence of unmet needs for limiting childbirth among WLHIV was assumed to be 50% with the precision of 5%, 20% refusal or incomplete data and the design effect was estimated at 1. Based on this assumption, the sample size was estimated at 461 WLHIV" [22].

## Data collection

Data was collected on a face-to-face basis among participants using a structured and pretested questionnaire in French, explained in the local language for participants if needed. For each participant, the questionnaire was filled by a trained health worker. This questionnaire included socio-demographic information, clinical features, information on ART, sexual activity status and on contraceptives knowledge and its use. Data on HIV status disclosure to the sexual partner was also collected. We defined unmet needs for limiting childbirth as the fecund women who were sexually active and intend to stop childbearing (limiting) but were not using any contraceptive method. We did not included needs for birth spacing.

## Data's statistical analysis

Data entry was performed using Epi Data software version 3.1. Data were then exported for statistical analyses using STATA/SE, version 15.1 (Statacorp LP, College Station, Texas, USA).

In descriptive analysis step, for continuous variables, mean and standard deviation were calculated while for categorical variables we calculated proportions. Our main outcome variable was WLHIV who reported unmet needs for limiting childbirth coded 1 and 0 if else. Pearson chi-square test or Fisher's exact test were used when appropriate in bivariate analysis All variables significant during bivariate analysis at a p-value <0.10 were included in a multivariate model.

A multivariable Poisson regression analysis was performed to identify independent factors associated with the dichotomous outcome "unmet needs for limiting childbirth or no". The estimates were presented as prevalence ratio (PR). All these analyses were performed with 95% confident interval (CI).

A multimodal averaging method was therefore performed using both a Poisson regression model (with robust variance) and the Akaike information criterion (AIC) for weighting models, based on the contribution of each covariate in explaining the risk of unmet needs for limiting childbirth. This process helps not only to select a final model (all the possible models) with Poisson regression procedures, but also to rank the covariates according to their relative importance. It compares the likelihood of an empty model with the likelihood of the model with covariates, provides also the proportion of the variation explained by the specified model. This averaging multimodel approach was described elsewhere [25].

In addition, we used relative importance weights (values between 0 and 1) to classify the associated factors according to the weight of the evidence with the following classification [26]: [0–0.5 [= no evidence; [0.5–0.75 [= weak evidence; [0.75–0.95 [= positive evidence; [0.95–0.99 [= strong evidence; [0.99–1 [= very strong evidence.

## Ethical issues

This study was approved by the National AIDS and STI Program of Togo (Ref N˚ 098/2016/MS/DSSP/PNLS-IST). We obtained consent from patients that participated in the study. For

each respondent, the objectives and benefits of participating in the survey and its conduct were clearly stated, as well as their right to interrupt the interview without justification. An informed consent form signed after the verbal explanation was made by the investigating officer in the language understood by the participant. For participants the aged between 15 and 17, we asked for the consent of the parents or the legal guardian.

## Results

### Socio-demographic and clinical characteristics

Table 1 shows the socio-demographic and clinical characteristics of the participants. In total 461 WLHIV of reproductive age and sexually active were enrolled into this study, we excluded 18 (3.9%) participants who were pregnant at the time of the survey. Of the 443 participants, 252 (56.9%) were living in the Centrale health region and 191 (43.1%) in the Kara health region. The mean ± (Standard Deviation) age of the participants was 34.5±7.0 years, ranging from 16 to 49 years, and half of them aged 35 years or older. Among participants, 40.9% had primary education level, and 45.1% had secondary or higher education level, 55.1% lived in couple and 62.5% lived in urban area. Of the 443 WLHIV who were interviewed, 403 (91.0%) were on ART and for more than two years for 71.4% of them. The mean duration on ART was 4.1 years ± 2.8 (SD). A quarter (25.7%) of participants had a CD4 cells count $< 350$ cells/mm$^3$ at the last visit. At the time of the survey, 52.4% of participants reported moderate HIV symptoms, while symptoms were severe in 4.5% of participants. The partner's HIV status was unknown for 42.4% (188/443) of participants. Most of the patients were followed-up in a public healthcare facility (60.9%), with a psychologist (56.7%). Three hundred and sixty-three participants had at least one child at the time of the survey.

### Unmet needs for limiting childbirth

Based on the conceptual definition in this study, 9.0% (95%CI [6.7–12.1]) of participants expressed unmet needs for limiting childbirth. However, this prevalence varies cross participants characteristics. Indeed, the proportion of participants with unmet needs was significantly lower (p = 0.004) among under 35 women than those aged from 35 years and older (5.0% vs 12.9%). A higher proportion of participants in couple (11.9%, p = 0.020), those having four children or more (17.0%, p = 0.037) and those who reported severe HIV symptoms (25.0%, p = 0.008) expressed also unmet needs for limiting childbirth during this study. However, WLHIV living in the Kara health region (p = 0.015), those followed-up in private healthcare facility (p = 0.001) or those followed-up in a center with a psychologist (p = 0.004) were less susceptible to report unmet needs for limiting childbirth (Table 1).

### Factors associated with unmet needs for limiting childbirth

After adjustment for significant covariates (with a p-value <0.10), multivariable analysis revealed a strong association between unmet needs for limiting childbirth, age of patients and the type of healthcare facility, a positive association with marital status and intensity of HIV symptoms and a weak association with the health region. In this study, older participants were more likely to report unmet needs for limiting childbirth than the younger. The prevalence of unmet needs for limiting childbirth was almost three times higher in WLHIV aged 35 or older (aPR = 3.11, 95%CI [1.52–6.38]; p = 0.002) than those aged less than 35 years. Concerning the marital status, the prevalence of unmet needs for limiting childbirth was two times higher among participants living in couple (aPR = 2.32 [1.15–4.65]; p = 0.018) compared to those who were single. Prevalence of unmet needs for limiting childbirth were 90% and 92% lower

**Table 1. Characteristics of the study participants (WLHI, N = 443).**

| | n (%) | Unmet needs | | p-value |
| --- | --- | --- | --- | --- |
| | | No, n (%) | Yes, n (%) | |
| **Age,** (mean±SD) years (443) | 34.5±7.0 | | | **0.004** |
| < 35 | 218 (49.2) | 207 (95.0) | 11 (5.0) | |
| 35–49 | 225 (50.8) | 196 (87.1) | 29 (12.9) | |
| **Education level** | | | | 0.161 |
| No education | 62 (14.0) | 58 (93.5) | 4 (6.5) | |
| Primary | 181 (40.9) | 159 (87.9) | 22 (12.1) | |
| Secondary and more | 200 (45.1) | 186 (93.0) | 14 (7.0) | |
| **Profession of participant** | | | | 0.187** |
| Public sector | 33 (7.4) | 31 (93.9) | 2 (6.1) | |
| Private sector | 19 (4.3) | 18 (94.7) | 1 (5.3) | |
| Informal sector | 226 (51.0) | 199 (88.1) | 27 (11.9) | |
| No profession | 165 (37.3) | 155 (93.9) | 10 (6.1) | |
| **Marital status** | | | | **0.020** |
| In couple | 244 (55.1) | 215 (88.1) | 29 (11.9) | |
| Single | 199 (44.9) | 188 (94.5) | 11 (5.5) | |
| **Residence of patient** | | | | 0.735 |
| Urban | 277 (62.5) | 251 (90.6) | 26 (9.4) | |
| Rural | 166 (37.5) | 152 (91.6) | 14 (8.4) | |
| **Region** | | | | **0.015** |
| Centrale | 252 (56.9) | 222 (88.1) | 30 (11.9) | |
| Kara | 191 (43.1) | 181 (94.8) | 10 (5.2) | |
| **Religion** | | | | 0.664** |
| None | 38 (8.6) | 36 (94.7) | 2 (5.3) | |
| Islam | 148 (33.4) | 135 (91.2) | 13 (8.8) | |
| Christianism | 257 (58.0) | 232 (90.3) | 25 (9.7) | |
| **WHO's clinical stage** | | | | **0.013** |
| Stage I | 243 (55.2) | 226 (93.0) | 17 (7.0) | |
| Stage II | 118 (26.8) | 109 (92.4) | 9 (7.6) | |
| Stage III & IV | 79 (18.0) | 65 (82.3) | 14 (17.7) | |
| **Symptoms' Intensity** | | | | **0.018** |
| None | 191 (43.1) | 170 (89.0) | 21 (11.0) | |
| Moderate | 232 (52.4) | 218 (94.0) | 14 (6.0) | |
| Severe | 20 (4.5) | 15 (75.0) | 5 (25.0) | |
| **CD4 cell counts** | | | | 0.789 |
| < 350 | 114 (25.7) | 103 (90.4) | 11 (9.7) | |
| ≥ 350 | 329 (74.3) | 300 (91.2) | 29 (8.8) | |
| **ART** | | | | 0.389 |
| Yes | 403 (91.0) | 368 (91.3) | 35 (8.7) | |
| No | 40 (9.0) | 35 (87.5) | 5 (12.5) | |
| **ART's scheme (n = 403)** | | | | 0.340* |
| 1rst line | 370 (91.8) | 336 (90.8) | 34 (9.2) | |
| 2nd line | 33 (8.2) | 32 (97.0) | 1 (3.0) | |
| **ART duration (= 414)** | | | | 0.669 |
| < 2 years | 115 (28.6) | 104 (90.4) | 11 (9.6) | |
| ≥2 years | 287 (71.4) | 263 (91.6) | 24 (8.4) | |
| **Partner's HIV status** | | | | 0.583 |

(*Continued*)

**Table 1.** (Continued)

| | n (%) | Unmet needs | | p-value |
| --- | --- | --- | --- | --- |
| | | No, n (%) | Yes, n (%) | |
| Unknown | 188 (42.4) | 174 (92.5) | 14 (7.5) | |
| HIV-positive | 134 (30.3) | 121 (90.3) | 13 (9.7) | |
| HIV-negative | 121 (27.3) | 108 (89.3) | 13 (10.7) | |
| **Type of health center** | | | | **<0.0001** |
| Private | 173 (39.1) | 168 (97.1) | 5 (2.9) | |
| Public | 270 (60.9) | 235 (87.0) | 35 (13.0) | |
| **Presence of psychologist in the center** | | | | **0.004** |
| No | 192 (43.3) | 166 (86.5) | 26 (13.5) | |
| Yes | 251 (56.7) | 237 (94.4) | 14 (5.6) | |
| **Number of children** | | | | **0.037** |
| None | 53 (12.7) | 51 (96.1) | 2 (4.7) | |
| 1–3 | 304 (73.1) | 279 (91.8) | 25 (8.2) | |
| 4–7 | 59 (14.2) | 49 (83.0) | 10 (17.0) | |
| **Fertility desire** | | | | **<0.0001** |
| No | 175 (40.0) | 135 (77.1) | 40 (22.9) | |
| Yes | 263 (60.0) | 263 (100.0) | 0 (0.0) | |
| **Contraceptive methods** | | | | **<0.0001** |
| None | 119 (26.9) | 80 (67.2) | 39 (32.7) | |
| Condom alone | 242 (54.6) | 241 (99.6) | 1 (0.4) | |
| Condom+hormonal contraceptive | 55 (12.4) | 55 (100.0) | 0 (0.0) | |
| Hormonal contraceptive | 26 (5.9) | 26 (100.0) | 0 (0.0) | |
| Intrauterine devices | 1 (0.2) | 1 (100.0) | 0 (0.0) | |

among patients respectively living in the Kara region (aPR = 0.10 [0.01–0.76]; p = 0.026) and followed in a private healthcare center (aPR = 0.08 [0.01–0.53]; p = 0.009) than respectively those living in the Centrale region and those who were followed in a public settings. Finally, WLHIV who reported severe HIV symptoms were more likely to express unmet needs for limiting childbirth (aPR: 3.50, 95%CI [1.31–9.37]; p = 0.013) than those without HIV symptoms (Table 2).

## Discussion

This cross-sectional study described an important public health issue in a country where reproductive health and the fight against HIV are the focus of health policy. It is one of the few studies that focused on reproductive health as well as unmet need for limiting childbirth among WLHIV in Togo. Indeed, 40% of participants expressed the desire not to have children any more in the future, but some of them did not use any contraceptive method. In our study, the overall prevalence of unmet need for limiting childbirth was estimated at 9%, which is lower than previously reported in the general population in Togo (12%) [24] as well as that reported in most epidemiological studies of conducted among WLHIV in Sub-Saharan Africa. In a cross-sectional studies among WLHIV, Yotebieng et al in DR Congo and Wanyenze et al in Uganda found respectively that 17.6% and 36.8% of the WLHIV had an unmet need for limiting childbirth [20,27]. However, similar result of 9% was reported in another study among WLHIV attending HIV Care and Treatment Service at Saint Paul's Hospital Millennium Medical College in Addis Ababa, Ethiopia [28]. Although the prevalence of unmet need for limiting childbirth was low among WLHIV in our study, these results nevertheless suggest that it is

**Table 2. Factors associated with unmet needs for limiting childbirth (multi-model averaging, N = 443).**

| | Poisson regression models | | | | | |
| | Univariate | | Multivariate | | Akaike weights (level of evidence) | Rank |
| | cPR | p-value | aPR | p-value | | |
|---|---|---|---|---|---|---|
| **Age,** years | | | | | | |
| < 35 | 1 | | 1 | | | |
| 35–49 | 2.55 [1.22–4.28] | **0.006** | 3.46 [1.76–6.82] | **<0.0001** | **0.99 (strong)** | **1** |
| **Education level** | | | | | | |
| No education | 1 | | | | | |
| Primary | 1.88 [0.67–5.26] | 0.227 | | | | |
| Secondary and more | 1.09 [0.37–3.20] | 0.882 | | | | |
| **Profession of participant** | | | | | | |
| No profession | 1 | | | | | |
| Public sector | 1.0 [0.23–4.36] | 1.0 | | | | |
| Private sector | 0.86 [0.12–6.43] | 0.890 | | | | |
| Informal sector | 1.97 [0.98–3.96] | 0.057 | | | | |
| **Marital status** | | | | | | |
| In couple | 2.15 [1.10–4.20] | **0.025** | 2.28 [1.23–4.24] | **0.009** | **0.87 (positive)** | **3** |
| Single | 1 | | 1 | | | |
| **Residence of patient** | | | | | | |
| Urban | 1 | | | | | |
| Rural | 0.90 [0.48–1.67] | 0.736 | | | | |
| **Region** | | | | | | |
| Centrale | 1 | | 1 | | | |
| Kara | 0.44 [0.22–0.88] | **0.020** | 0.10 [0.01–0.82] | **0.032** | **0.68 (weak)** | **5** |
| **Religion** | | | | | | |
| None | 1 | | | | | |
| Islam | 1.67 [0.39–7.09] | 0.488 | | | | |
| Christianism | 1.84 [0.46–7.50] | 0.390 | | | | |
| **Symptoms' Intensity** | | | | | | |
| None | 1 | | 1 | | | |
| Moderate/severe | 0.55 [0.29–1.05] | 0.070 | 0.70 [0.37–1.34] | 0.286 | **0.89 (positive)** | **4** |
| | 2.27 [0.96–5.38] | 0.061 | 4.39 [1.90–10.14] | **0.001** | | |
| **CD4 cell counts** | | | | | | |
| < 350 | 1 | | | | | |
| ≥ 350 | 0.91 [0.47–1.77] | 0.879 | | | | |
| **ART** | | | | | | |
| No | 1 | | | | | |
| Yes | 0.69 [0.29–1.67] | 0.417] | | | | |
| **ART's scheme (n = 415)** | | | | | | |
| 1rst line | 1 | | | | | |
| 2nd line | 0.33 [0.05–2.34] | 0.267 | | | | |
| **ART duration (= 414)** | | | | | | |
| < 2 years | 1 | | | | | |
| ≥2 years | 0.87 [0.44–1.73 | 0.699 | | | | |
| **Knowledge of partner's** | | 0.284 | | | | |
| **HIV status** | | | | | | |
| No | 1 | | | | | |
| Yes | 1.37 [0.73–2.55] | 0.323 | | | | |

*(Continued)*

**Table 2.** (Continued)

| | Poisson regression models | | | | | |
| | Univariate | | Multivariate | | Akaike weights (level of evidence) | Rank |
| | cPR | p-value | aPR | p-value | | |
|---|---|---|---|---|---|---|
| **Type of health center** | | | | | | |
| Public | 1 | | 1 | | | |
| Private | 0.22 [0.09–0.56] | **0.001** | 0.07 [0.01–0.48] | **0.007** | **0.98 (strong)** | **2** |
| **Presence of psychologist in the center** | | | | | | |
| No | 1 | | 1 | | | |
| Yes | 0.41 [0.22–0.77] | **0.005** | 8.51 [0.98–73.85] | 0.051 | **0.59 (weak)** | **6** |
| **Number of children** | | | | | | |
| None | 1 | | | | | |
| 1–3 | 2.18 [0.53–8.95] | 0.280 | | | | 7 |
| 4–7 | 4.49 [1.03–19.61] | **0.046** | | | | |

According to the value of the importance weights [26]:

[0–0.5]: no evidence.

[0.5–0.75]: weak evidence.

[0.75–0.90]: positive evidence.

[0.95–0.99]: strong evidence.

[0.99–1]: very strong evidence.

Based on importance weight.

very important to implement strategies to address these needs, thereby reducing the risk of MTCT of HIV. To effectively guide the development of these strategies at the level of the study regions, the results of this study among WLHIV allow us to rank the factors associated with unmet need for limiting childbirth according to the importance of weight.

This study found that the age of the participants was the main factor that could influence mostly the prevalence of unmet need for limiting childbirth, with older WLHIV being more likely to have an unmet need to limit childbirth. This is consistent with the findings of a study conducted in HIV clinics at Mulago in Uganda, where WLHIV were about six times more likely to have unmet need for FP [20]. Older WLHIV may probably experience the side effects of contraception use and might be reluctant to use modern contraceptive. In addition, older women may have a lower perception of the risk of pregnancy.

WLHIV attending private HIV clinics were less likely to have unmet needs for limiting childbirth compared to those attending public HIV clinics. Private HIV clinics may provide better reproductive health services, including FP services. They are most often highly motivated to improve client satisfaction by providing integrated reproductive health care, and also work better at retaining patients living with HIV [29]. This result was consistent with that found in a study done in Mexico which reported that women were less likely to have unmet needs for limiting if they had access to private health service [30].

In this study, WLHIV in couple, either married or not, were more likely to report unmet need for limiting childbirth compared to those who were sexually active but not in couple. This result was consistent with studies conducted in Ethiopia [28] which found that WLHIV who were married were higher to report unmet need for limiting childbirth. In addition, similar findings from a study in Zambia and Swaziland [31] reported that WLHIV living in couples were twice as likely to have an unmet need for limiting childbirth. These women may have already the number of children that they wanted and may express their willingness to limit childbearing.

Not surprisingly, WLHIV who have reported moderate or severe symptoms of AIDS are more likely to have an unmet need for limiting childbirth. Indeed, in general, WLHIV who have symptoms of AIDS most often make the priority choice to regain their well-being and relegate to the second plan other health care, including reproductive health care. But WLHIV receiving antiretroviral therapy become more sexually active which could be accompanied by return to fertility [32]. In that context, most of them are reluctant to be pregnant, fearing transmit HIV to their child. In LMIC, where reproductive health services are not often integrated in the basic care in the HIV clinics, WLHIV may have to their demand for reproductive health services not satisfied [33]. However, those with moderate or severe symptoms may need to take additional pills each day for eventually treatment of opportunistic infections or symptomatic relief. Fearing potential drug interactions, clinician delay the use of contraceptive methods, hormonal contraceptives, as much as possible [34].

Finally, it should be noted that there is a disparity of the prevalence of unmet need for limiting childbirth between health regions. We found that WLHIV living in Kara health region were less likely to have unmet need for limiting childbirth compared to those living in Centrale health region. This variability between these two regions could be explained not only by cultural differences but also by the poverty index in these regions. Indeed, the central region is one of the poorest regions of Togo [35] and this could mostly influence the use of modern contraceptive. Similarly, a study conducted in Mexico using national demographic survey found that women living in the poorest region of the country have greater unmet needs for both spacing and limiting than those living in the capital of the country [30].

This study assessed the relative weight of each risk factor on unmet need for limiting childbirth among WLHIV and showed that the women's age and the type of HIV clinic greatly influenced the prevalence of the unmet need for limiting childbirth. This suggests that, to be effective, interventions to eradicate unmet need for limiting childbirth should target WLHIV aged 35 years and older or those attending public HIV clinics in Togo.

## Limitations of study

This study has several limitations. It was conducted in health facilities with a low probability to include WLHIV who were not followed-up or less regularly followed-up in the HIV clinics including in this study. They may have different characteristics. As for with most cross-sectional studies, not only the study sample may not be representative of all the WLHIV residing in Togo, but also it should be noted a possible reverse causality.

## Conclusion

This research showed that the prevalence of unmet need for limiting childbirth among WLHIV was relatively low, but heterogeneous across the health regions involved in this study. Older women (aged 35 years and older), those in couple, those attending public HIV clinics, those with moderate or severe symptoms of AIDS, and those residing in the Centrale health region had the highest unmet needs for contraception to limit childbirth. Interventions to improve more access to modern contraception, targeting women aged 35 to 49 years, those in couple or those attending public healthcare facilities would contribute to the eradication of mother-to-child transmission of HIV in Togo.

## Supporting information

**S1 Data.**
(PDF)

**S2 Data.**
(PDF)

## Acknowledgments

We would like to thank health workers involved in data collection for their contribution. We acknowledge all patients who accepted to participate in this study.

## Author Contributions

**Conceptualization:** Issifou Yaya, Akouda Akessiwe Patassi.

**Data curation:** Issifou Yaya, Sokhna Dieng, Lihanimpo Djalogue, Mawuényégan Kouamivi Agboyibor, Mathias Kouamé N'Dri, Takiyatou Baba-Toherou, Akouda Akessiwe Patassi, Dadja Essoya Landoh, Kanfitine Kolani, Abdoul-Samadou Aboubakari.

**Formal analysis:** Issifou Yaya, Aboubakari Nambiema, Mawuényégan Kouamivi Agboyibor, Mathias Kouamé N'Dri.

**Investigation:** Issifou Yaya.

**Methodology:** Issifou Yaya, Sokhna Dieng, Bayaki Saka.

**Supervision:** Kanfitine Kolani, Bayaki Saka.

**Validation:** Lihanimpo Djalogue, Mathias Kouamé N'Dri, Takiyatou Baba-Toherou, Akouda Akessiwe Patassi, Dadja Essoya Landoh, Kanfitine Kolani, Abdoul-Samadou Aboubakari, Bayaki Saka.

**Visualization:** Takiyatou Baba-Toherou, Akouda Akessiwe Patassi.

**Writing – original draft:** Issifou Yaya, Aboubakari Nambiema, Sokhna Dieng, Lihanimpo Djalogue, Mawuényégan Kouamivi Agboyibor, Mathias Kouamé N'Dri, Takiyatou Baba-Toherou, Akouda Akessiwe Patassi, Dadja Essoya Landoh, Kanfitine Kolani, Abdoul-Samadou Aboubakari, Bayaki Saka.

**Writing – review & editing:** Issifou Yaya, Aboubakari Nambiema, Sokhna Dieng, Lihanimpo Djalogue, Bayaki Saka.

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
