## [Decision Letter · Decision Letter 0]

26 Feb 2020

PONE-D-20-01103

Factors associated with unmet need for limiting childbirth among HIV-positive women in Togo: an averaging approach

PLOS ONE

Dear Dr Saka,

Thank you for submitting your manuscript to PLOS ONE. After careful consideration, we feel that it has merit but does not fully meet PLOS ONE’s publication criteria as it currently stands. Therefore, we invite you to submit a revised version of the manuscript that addresses the points raised during the review process.

We would appreciate receiving your revised manuscript by Apr 11 2020 11:59PM. To enhance the reproducibility of your results, we recommend that if applicable you deposit your laboratory protocols in protocols.io, where a protocol can be assigned its own identifier (DOI) such that it can be cited independently in the future. For instructions see: http://journals.plos.org/plosone/s/submission-guidelines#loc-laboratory-protocols

We look forward to receiving your revised manuscript.

Kind regards,

Kwasi Torpey, MD PhD MPH

Academic Editor

PLOS ONE

Journal Requirements:

2. Please include additional information regarding the survey or questionnaire used in the study and ensure that you have provided sufficient details that others could replicate the analyses. If you developed and/or translated a questionnaire as part of this study and it is not under a copyright more restrictive than CC-BY, please include a copy, in both the original language and English, as Supporting Information.

3. We noticed you have some minor occurrence(s) of overlapping text with the following previous publication(s), which needs to be addressed:

https://doi.org/10.1136/bmjopen-2017-019006

In your revision ensure you cite all your sources (including your own works), and quote or rephrase any duplicated text outside the Methods section. Further consideration is dependent on these concerns being addressed.

5. We note you have included a table to which you do not refer in the text of your manuscript. Please ensure that you refer to Table 2 in your text; if accepted, production will need this reference to link the reader to the Table.

Reviewers' comments:

Reviewer's Responses to Questions

**Comments to the Author**

1. Is the manuscript technically sound, and do the data support the conclusions?

Reviewer #1: Partly

Reviewer #2: Yes

2. Has the statistical analysis been performed appropriately and rigorously? 

Reviewer #1: No

Reviewer #2: Yes

3. Have the authors made all data underlying the findings in their manuscript fully available?

Reviewer #1: Yes

Reviewer #2: Yes

4. Is the manuscript presented in an intelligible fashion and written in standard English?

Reviewer #1: Yes

Reviewer #2: Yes

5. Review Comments to the Author

Reviewer #1: Understanding unmet FP needs of women with HIV+ women is important to understand programmatic gaps, meet women's RH goals, and reduce MTCT. The authors present data from women in Togo on this topic, and identify some gaps. There are some notable methodological issues (namely in the definition of the outcome and in analysis/lack of accounting for clustering) that are important to fully comprehend study results. In addition, the authors could do a better job in the background and discussion describing the rationale for the study (ie many studies have already been conducted, although not in Togo) and synthesize findings in the discussion rather than only repeat findings and compare to the literature.

Abstract and Methods

-Unmet need – did you exclude needs for birth spacing? If so please overtly state in methods. If not, add birth spacing to definition of unmet need.

Background

-The background could be more focused; contraception isn’t mentioned until the 3rd paragraph. Suggest cutting much of 1st 3-4 paragraphs to narrow focus on your topic.

Methods

-The sample was selected at the facility level and then participants within the facility. The authors should account for clustering in all of their analysis.

-Describe the rationale for the AIC/Poisson model; ie how it differs from the multivariate model and why it was done.

-In the analysis of unmet need who is the comparator, women with their FP needs met who desire to not have future children? Or are woman who want to conceive in the comparator? This is an important methodological issue, as if women who want to conceive are in the comparison group than there is a lot of heterogeneity in the comparison group (needs met + desire for future children); these groups are very different.

Results

-Results need more specificity in their description; for example age of patient and type of healthcare facility are associate with unmet need. These are variables but don’t describe the association (ie older age, or younger age?). Suggest cutting these types of statements as later you go on to describe the relationship with age > and < 35.

-“Weak with the health region”unclear sentence.

- Was contraceptive use measured? Among women who had their “needs met” what was the method mix? Women who are not using condoms plus another method (dual method) technically have their FP needs met, with a less effective method. In contrast, women using DMPA have FP needs met but are at risk for HIV transmission and STIs. Would be nice to more comprehensively describe the cohort.

-Did the authors look to see if any of their variables were collinear in the MV model? Ie age and # children?

Discussion

-The authors did not “address” a public health issue as it is just descriptive; suggest rephrasing.

-There are actually several studies on RH and unmet need (the authors cited some in the background but there are many more). Claiming this sis one of a few studies is not accurate.

-The comparison with the Uganda results does not make sense; you are discussing age among HIV+ women whereas the Uganda study was comparing HIV+ to HIV-. This should be removed for comparison or change context to refer to % unmet need among HIV+.

-The marital status findings are mentioned 2X in the discussion, suggest consolidating into one section.

-The interpretation of the women with severe findings is inconsistent with your results. You found women with symptoms were MORE likely to have unmet need, but the following sentence supports these women having LESS unmet need. Please revise accordingly.

Limitations

-You should acknowledge possibility of reverse causality due to the cross-sectional design.

Conclusions

-This seems to repeat the study findings but should describe the “so what” about the research. What are next steps as a result of this research?

Table 1

-Unmet need should be defined, including making it clear who is in the “no” category.

-N for age is not listed

-Can you include partner status, would be helpful to know among those with known partner status if partners are HIV+ vs HIV-

-Why is fertility desire not described for both groups?

-Unmet need should not be in the rows it is the outcome.

The manuscript would benefit from a native English speaker review/copyedit. There are several grammatical errors throughout.

Reviewer #2: Overall comment:

This is an interesting cross-sectional study that examined the prevalence of unmet need for limiting childbirth and associated factors among women living with HIV in Togo. This is an important and timely piece of work conducted at a time when the global community is increasing efforts to close the remaining gaps in prevention of mother-to-child transmission of HIV programs.

While the study is analytically sound and clearly written, there was some lack of detail and consistency regarding the definition of the outcome and some key variables. The authors should expand the Methods section to include details of appropriate definitions of the outcome and key variables, including questions from which these variables were constructed. Also, I would advise the authors to revise extensively for typos to improve readability of the text.

Abstract and Background:

1. The terms ‘in a relationship’, ‘marital status’ ‘in a couple’ mean different things and have been used interchangeably in the abstract (result and conclusion) and also throughout the paper. The definition of relationship status e.g. cohabiting (married or not) should be clarified and consistently used.

2. Depending on editorial preference, consider using women living with HIV rather than HIV-positive women.

3. Page 3: first sentence in last paragraph is unclear. Rephrase sentence.

4. Page 3, second paragraph line 9: uncap the ‘Sub’ in ‘Sub-Saharan’

Methods:

5. Similar to the varying prevalence of unmet need for limiting childbirth across health regions, I wonder if you could add to your predictors some metric (if you have this in your database) for patients’ ease of accessibility to health facility? For instance, distance to the clinic?

6. Page 6: line 1 and 2 implies a contradiction to the earlier definition of your study population. My understanding based on the inclusion criteria on page 5 is that all women included were sexually active. Clarify.

7. Page 7, first paragraph: Clearly define how you categorized ‘moderate vs. severe’ HIV symptoms.

Results:

8. I wonder if some information on type of contraceptive methods used among those without the outcome can be included. It would be useful to better understand possible targeted interventions to improve unmet needs in the region.

9. For the categorical variable that you created for age, is there a reason why this was not a 3-level variable? Considering the age range of 16-49 in your study sample, I imagine that fertility intentions, contraceptive use and health seeking behaviors might be different for 16-25 year old women compared to those 26-35 years of age.

Discussion:

10. I had some concerns about the potential for under representation of unmet needs for limiting childbirth in your sample, due to the fact that you recruited women who were followed at HIV care sites for at least 6 months, but I think you have addressed this adequately in the limitations.

11. Page 11, Line 2 in Conclusion: a word is missing between ‘health’ and ‘involved’

6. PLOS authors have the option to publish the peer review history of their article (what does this mean?). If published, this will include your full peer review and any attached files.

Reviewer #1: No

Reviewer #2: No

---

## [Author Response · Author response to Decision Letter 0]

16 Apr 2020

Authors‘ response

Here is a point-by-point response to the Editor and the reviewers’ comments and concerns.

Editor Comments:

Answer: As suggested the editor, we have formatted our manuscript as required. 

2. Please include additional information regarding the survey or questionnaire used in the study and ensure that you have provided sufficient details that others could replicate the analyses. If you developed and/or translated a questionnaire as part of this study and it is not under a copyright more restrictive than CC-BY, please include a copy, in both the original language and English, as Supporting Information.

Answer: Thank you so much, As suggested the editor, we have included the French version of the questionnaire used in the study. We are still working to translate it in English.

3. We noticed you have some minor occurrence(s) of overlapping text with the following previous publication(s), which needs to be addressed: https://doi.org/10.1136/bmjopen-2017-019006

In your revision ensure you cite all your sources (including your own works), and quote or rephrase any duplicated text outside the Methods section. Further consideration is dependent on these concerns being addressed.

Answer: Thank you so much. As suggested the editor, we have cited my previous paper and put overlapping text in quote.

Answer: Here is my ORCID iD : orcid.org/0000-0002-2554-6515

5. We note you have included a table to which you do not refer in the text of your manuscript. Please ensure that you refer to Table 2 in your text; if accepted, production will need this reference to link the reader to the Table.

Answer: Thank you for this comment, we included the reference of the table 2 as it is required in the text of our manuscript

 

Reviewer #1: Understanding unmet FP needs of women with HIV+ women is important to understand programmatic gaps, meet women's RH goals, and reduce MTCT. The authors present data from women in Togo on this topic, and identify some gaps. There are some notable methodological issues (namely in the definition of the outcome and in analysis/lack of accounting for clustering) that are important to fully comprehend study results. In addition, the authors could do a better job in the background and discussion describing the rationale for the study (ie many studies have already been conducted, although not in Togo) and synthesize findings in the discussion rather than only repeat findings and compare to the literature.

We would like to thank the reviewer for careful reading of our manuscript and for the valuable comments, which were of great help in revising in order to improve the quality of this manuscript. Our responses to the reviewer’s comments are given below.

Abstract and Methods

-Unmet need – did you exclude needs for birth spacing? If so please overtly state in methods. If not, add birth spacing to definition of unmet need.

Answer: Thank you for this comment, we did not included needs for birth spacing in the outcome definition (Unmet need for limiting childbirth), so we stated it in the methods, thank you

Background

-The background could be more focused; contraception isn’t mentioned until the 3rd paragraph. Suggest cutting much of 1st 3-4 paragraphs to narrow focus on your topic.

Answer: As suggested the reviewer, we have deleted the first paragraph and we have reorganized the other paragraphs by focusing more on family planning and contraceptive use, thank you

Methods

-The sample was selected at the facility level and then participants within the facility. The authors should account for clustering in all of their analysis.

Answer: Thank you for this comment. In our statistical analysis strategies, we had tried multilevel regression models taking into account the health facilities. However, the variability between health facilities was not significant. So, we had to abandon this analysis strategies.

-Describe the rationale for the AIC/Poisson model; ie how it differs from the multivariate model and why it was done.

Answer: As suggested the reviewer, we describe with more details the rationale in the methods section. So we added in the “Data’s statistical analysis” section the following paragraph: “This process helps not only to select a final model (all the possible models) with Poisson regression procedures, but also to rank the covariates according to their relative importance. It compares the likelihood of an empty model with the likelihood of the model with covariates, provides also the proportion of the variation explained by the specified model. This averaging multimodel approach was described elsewhere [25]”

The multivariable model provided adjusted prevalence ratio, while multimodal averaging helped to rank covariates based on their contribution in explaining the risk of unmet needs for limiting childbirth.

-In the analysis of unmet need who is the comparator, women with their FP needs met who desire to not have future children? Or are woman who want to conceive in the comparator? This is an important methodological issue, as if women who want to conceive are in the comparison group than there is a lot of heterogeneity in the comparison group (needs met + desire for future children); these groups are very different.

Answer: Thank you for this relevant comment. We agree with the reviewer that the comparison group in our study is very heterogeneous. 

Results

-Results need more specificity in their description; for example, age of patient and type of healthcare facility are associate with unmet need. These are variables but don’t describe the association (ie older age, or younger age?). Suggest cutting these types of statements as later you go on to describe the relationship with age > and < 35.

-“Weak with the health region”unclear sentence.

Answer: Thank you for this comment. We correct it, thank you

- Was contraceptive use measured? Among women who had their “needs met” what was the method mix? Women who are not using condoms plus another method (dual method) technically have their FP needs met, with a less effective method. In contrast, women using DMPA have FP needs met but are at risk for HIV transmission and STIs. Would be nice to more comprehensively describe the cohort.

Answer: Thank you for pointing this out. We added in the Table 1 the description of type of contraception reported by participants. We agree with your comment, that not all methods protect against STIs. But as you will note, about two thirds of participants used condoms and could therefore avoid STIs. 

-Did the authors look to see if any of their variables were collinear in the MV model? Ie age and # children?

Answer: Thank you for this suggestion. We finally excluded the number of children in the MV model, as these variables are collinear.

Discussion

-The authors did not “address” a public health issue as it is just descriptive; suggest rephrasing.

Answer: we correct it, thank you

-There are actually several studies on RH and unmet need (the authors cited some in the background but there are many more). Claiming this sis one of a few studies is not accurate.

Answer: We agree with this comment. We correct it, we specified that is was related to studies conducted in Togo, we omitted to mention it, thank you

-The comparison with the Uganda results does not make sense; you are discussing age among HIV+ women whereas the Uganda study was comparing HIV+ to HIV-. This should be removed for comparison or change context to refer to % unmet need among HIV+.

Answer: As suggested the reviewer, we have removed this reference, thank you

-The marital status findings are mentioned 2X in the discussion, suggest consolidating into one section.

-The interpretation of the women with severe findings is inconsistent with your results. You found women with symptoms were MORE likely to have unmet need, but the following sentence supports these women having LESS unmet need. Please revise accordingly.

Answer: The reviewer has raised an important point here. So, we revised this paragraph in the “Discussion” section as following “Not surprisingly, WLHIV who have reported moderate or severe symptoms of AIDS are more likely to have an unmet need for limiting childbirth. Indeed, in general, WLHIV who have symptoms of AIDS most often make the priority choice to regain their well-being and relegate to the second plan other health care, including reproductive health care. But WLHIV receiving antiretroviral therapy become more sexually active which could be accompanied by return to fertility [32]. In that context, most of them are reluctant to be pregnant, fearing transmit HIV to their child. In LMIC, where reproductive health services are not often integrated in the basic care in the HIV clinics, WLHIV may have to their demand for reproductive health services not satisfied [33]. However, those with moderate or severe symptoms may need to take additional pills each day for eventually treatment of opportunistic infections or symptomatic relief. Fearing potential drug interactions, clinician delay the use of contraceptive methods, hormonal contraceptives, as much as possible [34]”.

Limitations

-You should acknowledge possibility of reverse causality due to the cross-sectional design.

Answer: As suggested the reviewer, we have added a sentence mentioning a possible reverse causality due to the cross-sectional design.

Conclusions

-This seems to repeat the study findings but should describe the “so what” about the research. What are next steps as a result of this research?

Answer: Thank you for this comment. We will disseminate our results to policy decision-makers in order firstly, to convince them to reproduce this study in other regions and secondly translate these results into effective interventions adapted to reduce unmet needs.

Table 1

-Unmet need should be defined, including making it clear who is in the “no” category.

Answer: Thank you for pointing this out. We defined unmet needs for limiting childbirth as the fecund women who were sexually active and intend to stop childbearing (limiting) but were not using any contraceptive method. And furthermore, our main outcome variable was WLHIV who reported unmet needs for limiting childbirth coded 1 (oui) and 0 (non) if else.

-N for age is not listed

Answer: As suggested the reviewer, we included it in the table 1, thank you

-Can you include partner status, would be helpful to know among those with known partner status if partners are HIV+ vs HIV-

Answer: As suggested the reviewer, we included it in the table 1, thank you

-Why is fertility desire not described for both groups?

Answer: the fertility desire is included in the outcome definition, it is why we decided not to include it in the description, thank you

-Unmet need should not be in the rows it is the outcome.

Answer: As suggested the reviewer, we deleted the row containing the outcome, thank you

The manuscript would benefit from a native English speaker review/copyedit. There are several grammatical errors throughout.

 

Reviewer #2: Overall comment:

This is an interesting cross-sectional study that examined the prevalence of unmet need for limiting childbirth and associated factors among women living with HIV in Togo. This is an important and timely piece of work conducted at a time when the global community is increasing efforts to close the remaining gaps in prevention of mother-to-child transmission of HIV programs.

While the study is analytically sound and clearly written, there was some lack of detail and consistency regarding the definition of the outcome and some key variables. The authors should expand the Methods section to include details of appropriate definitions of the outcome and key variables, including questions from which these variables were constructed. Also, I would advise the authors to revise extensively for typos to improve readability of the text.

Abstract and Background:

1. The terms ‘in a relationship’, ‘marital status’ ‘in a couple’ mean different things and have been used interchangeably in the abstract (result and conclusion) and also throughout the paper. The definition of relationship status e.g. cohabiting (married or not) should be clarified and consistently used.

Answer : Thank you for this comment. In our study, we have chosen to use the term “In couple” which was define as participant cohabiting with his/her sexual partner. We change it through the text of the manuscript.

2. Depending on editorial preference, consider using women living with HIV rather than HIV-positive women.

Answer : As suggested the reviewer, we have changed it through the whole manuscript

3. Page 3: first sentence in last paragraph is unclear. Rephrase sentence.

Answer : Thank you for this suggestion, we have rephrased the sentence.

4. Page 3, second paragraph line 9: uncap the ‘Sub’ in ‘Sub-Saharan’

Answer : As suggested the reviewer, we have corrected it 

Methods:

5. Similar to the varying prevalence of unmet need for limiting childbirth across health regions, I wonder if you could add to your predictors some metric (if you have this in your database) for patients’ ease of accessibility to health facility? For instance, distance to the clinic?

Answer: Thank so much for this suggestion, we agree the reviewer, these factors could help to improve the explanation of the outcome, but unfortunately, we did not collect them. 

6. Page 6: line 1 and 2 implies a contradiction to the earlier definition of your study population. My understanding based on the inclusion criteria on page 5 is that all women included were sexually active. Clarify.

Answer: We agree with this comment, we have corrected it, thank you

7. Page 7, first paragraph: Clearly define how you categorized ‘moderate vs. severe’ HIV symptoms.

Answer: Thank you for pointing this out. In our study, to provide information on the intensity of the symptoms; participants were asked to describe the symptoms of HIV through the following question: How do you describe your current HIV symptoms? and we proposed a response with 5 modalities :None = 1; medium = 2; moderate = 3; severe = 4; don't want to answer = 99. In the data analysis, we had combined the modalities 2 and 3 which takes the name of "moderate".

Results:

8. I wonder if some information on type of contraceptive methods used among those without the outcome can be included. It would be useful to better understand possible targeted interventions to improve unmet needs in the region.

Answer: As suggested the reviewer, we have added in the Table 1 the description of type of contraception reported by participants. Thank you.

9. For the categorical variable that you created for age, is there a reason why this was not a 3-level variable? Considering the age range of 16-49 in your study sample, I imagine that fertility intentions, contraceptive use and health seeking behaviors might be different for 16-25 years old women compared to those 26-35 years of age.

Answer: We agree with these comments. Initially, we create a variable with 3 levels <25; 26-35 and >35 years. However, in the regression models, we found that PR for 16-25 years old women was not statistically different compared to those 26-35 years of age. So, we decided to combine these two categories. Thank you.

Discussion:

10. I had some concerns about the potential for under representation of unmet needs for limiting childbirth in your sample, due to the fact that you recruited women who were followed at HIV care sites for at least 6 months, but I think you have addressed this adequately in the limitations.

Answer: Thank you for mention this.

11. Page 11, Line 2 in Conclusion: a word is missing between ‘health’ and ‘involved’

Answer: we correct it, thank you

---

## [Editor Report · Decision Letter 1]

24 Apr 2020

PONE-D-20-01103R1

Factors associated with unmet need for limiting childbirth among women living with HIV in Togo: an averaging approach

PLOS ONE

Dear Dr Saka,

Thank you for submitting your manuscript to PLOS ONE. After careful consideration, we feel that it has merit but does not fully meet PLOS ONE’s publication criteria as it currently stands. Therefore, we invite you to submit a revised version of the manuscript that addresses the points raised during the review process.

We would appreciate receiving your revised manuscript by 15th May 2020. To enhance the reproducibility of your results, we recommend that if applicable you deposit your laboratory protocols in protocols.io, where a protocol can be assigned its own identifier (DOI) such that it can be cited independently in the future. For instructions see: http://journals.plos.org/plosone/s/submission-guidelines#loc-laboratory-protocols

We look forward to receiving your revised manuscript.

Kind regards,

Professor Kwasi Torpey, MD PhD MPH

Academic Editor

PLOS ONE

Additional Editor Comments (if provided):

Thanks. The revised manuscript has improved. However, there are some comments that were not fully addressed

1. Reviewer 1: The comment below by Reviewer 1 and the corresponding response does not address the issue raised. Kindly ensure it is adequately addressed

-In the analysis of unmet need who is the comparator, women with their FP needs met who desire to not have future children? Or are woman who want to conceive in the comparator? This is an important methodological issue, as if women who want to conceive are in the comparison group than there is a lot of heterogeneity in the comparison group (needs met + desire for future children); these groups are very different.

Answer: Thank you for this relevant comment. We agree with the reviewer that the comparison group in our study is very heterogeneous

2. Copy-editing: The manuscript needs copyediting by native English speaker to address language corrections starting from the abstract then the main paper. See a few a examples (CAPS) in the abstract

Background: With the LARGE access to antiretroviral treatment...….. the subsequent sentence also needs revision

Methods: A cross-sectional study was conducted between June and August 2016, INCLUDING WLHIV of reproductive age (15–49 years), sexually active and followed-up in HIV-care settings in Centrale and Kara regions, in Togo. Data WERE collected on a face-to-face basis - The whole methods section in the abstract needs revision

---

## [Author Response · Author response to Decision Letter 1]

27 Apr 2020

"-In the analysis of unmet need who is the comparator, women with their FP needs met who desire to not have future children? Or are woman who want to conceive in the comparator? This is an important methodological issue, as if women who want to conceive are in the comparison group than there is a lot of heterogeneity in the comparison group (needs met + desire for future children); these groups are very different.

Answer: Thank you for this relevant comment. We agree with the reviewer that the comparison group in our study is very heterogeneous. In the comparator group we have women with their FP needs met who desire to not have future children and those who want to conceive."

Copy-editing: The manuscript needs copyediting by native English speaker to address language corrections starting from the abstract then the main paper. See a few a examples (CAPS) in the abstract

Answer: Done

Background: With the LARGE access to antiretroviral treatment...….. the subsequent sentence also needs revision

Answer: Done

Methods: A cross-sectional study was conducted between June and August 2016, INCLUDING WLHIV of reproductive age (15–49 years), sexually active and followed-up in HIV-care settings in Centrale and Kara regions, in Togo. Data WERE collected on a face-to-face basis - The whole methods section in the abstract needs revision

Answer: Done

---

## [Editor Report · Decision Letter 2]

29 Apr 2020

Factors associated with unmet need for limiting childbirth among women living with HIV in Togo: an averaging approach

PONE-D-20-01103R2

Dear Dr. Saka,

We are pleased to inform you that your manuscript has been judged scientifically suitable for publication and will be formally accepted for publication once it complies with all outstanding technical requirements.

With kind regards,

Professor Kwasi Torpey, MD PhD MPH

Academic Editor

PLOS ONE
---

## [Editor Report · Acceptance letter]

8 May 2020

PONE-D-20-01103R2 

Factors associated with unmet need for limiting childbirth among women living with HIV in Togo: an averaging approach 

Dear Dr. Saka:

I am pleased to inform you that your manuscript has been deemed suitable for publication in PLOS ONE. Congratulations! Your manuscript is now with our production department. 

With kind regards,

on behalf of

Professor Kwasi Torpey 

Academic Editor

PLOS ONE